# Mental Health during COVID-19 Pandemic: Qualitative Perceptions among Lithuanian Adolescents

**DOI:** 10.3390/ijerph19127086

**Published:** 2022-06-09

**Authors:** Justė Lukoševičiūtė, Kastytis Šmigelskas

**Affiliations:** 1Health Research Institute, Faculty of Public Health, Medical Academy, Lithuanian University of Health Sciences, Tilžės g. 18, 47181 Kaunas, Lithuania; kastytis.smigelskas@lsmu.lt; 2Department of Health Psychology, Faculty of Public Health, Medical Academy, Lithuanian University of Health Sciences, Tilžės g. 18, 47181 Kaunas, Lithuania

**Keywords:** adolescent, mental health, COVID-19, qualitative research

## Abstract

Background: Since the COVID-19 pandemic emerged, humanity has had to face unprecedented change in daily routines. Therefore, the pandemic has also had an impact on mental health. Most of the literature analyzes adult experiences during the COVID-19 pandemic, while the youth is less investigated. The purpose of this study was to reveal adolescent experiences during COVID-19. Methods: This qualitative study consisted of 19 adolescents from 11–17 years old. Semi-structured in-depth interviews were conducted, and thematic analysis was applied. Results: Five themes were identified: ambivalent feelings; daily routine changes; disappointment with distance education; coping strategies; and discoveries. Conclusion: The study revealed adolescents’ mostly negative feelings such as sadness, anger, loneliness, or boredom. They shared about frustrating daily routine changes and shifting to a distance education mode that was quite limited in effectiveness and convenience. Nonetheless, communication with family, peers, pets, active leisure, and favorite activities helped them to cope with the difficulties during COVID-19. During this period, adolescents had more free time for themselves and personal growth, found new activities, or improved some relationships. Overall, the COVID-19 pandemic had mostly negative side effects, and regardless of positive experiences, it was mainly considered by adolescents as an adversity for their mental health.

## 1. Introduction

An outbreak of coronavirus disease 2019 (COVID-19) started in late 2019 and was declared a pandemic by the World Health Organization on 11 March 2020 [1]. Since then, humanity has had to face unprecedented changes in the way daily routines are organized. The changes have affected work, social, and personal life. In many countries of the world work, business, traveling, and school activities were suspended. People were encouraged or forced to stay home. There were many uncertainties about this disease, with a possibility of a fatal outcome. Therefore, COVID-19 had a significant impact on mental health. The scientific literature showed increased levels of distress, anxiety, depression, and insomnia in general populations during this pandemic [2,3,4]. A systematic review and meta-analysis found that almost one-third of the general population in Asia and Europe suffered from stress (30%), anxiety (32%), or depression (34%) [5].

Adolescents were also affected, especially due to abrupt withdrawal from school, social life, and outdoor activities. Although epidemiological data show that children and adolescents are minimally susceptible to COVID-19 (based on rates of cases, hospitalization, and death in different ages groups) [6], they are hit hardest by the psychosocial impact of this pandemic: being in lockdown may impose a greater psychological burden than the physical suffering caused by the virus [7]. Children are more vulnerable also because of their limited understanding of the situation. They are unable to escape the physical and mental harms of such a situation, as they have limited coping strategies. It can also be assumed that children may not be able to communicate their feelings like adults [8]. De Figueiredo and colleagues [9] showed that some children experienced an increase in domestic violence. The stress they are subjected to directly impacts their mental health through increased anxiety, changes in diet and school dynamics, and fear or failure to scale the problem. Moreover, the systematic review by Panchal et al. [10] revealed risk and protective factors for adolescents’ poor mental health. Factors such as lack of routine, female gender, adolescence, excessive exposure to COVID information and media, previous mental health problems, community case frequency, and relative doing first-line jobs related to COVID-19 were related to poorer children’s and adolescents’ mental health. Meanwhile, all types of routine, family communication, social support, and appropriate play and leisure were defined as protective factors for adolescents’ good mental health. Therefore, in describing the COVID-19 pandemic the researchers found biological, environmental, and social factors playing a role in adolescent life.

Similar to researchers, mental health professionals also observed changes in adolescent mental health and behaviors. Most research has been conducted on mood and its disorders. A longitudinal study in Australia comparing pre-pandemic and first COVID-19 wave results showed that adolescents reported a significant increase in depressive symptoms and anxiety, alongside a decrease in life satisfaction [11]. An increase in depression at follow-up was associated with COVID-19-related worries, online learning difficulties, and increased conflict with parents. An increase in anxiety was associated with female gender, COVID-19 distress, media use, and social disconnection [11]. Other studies also confirm increasing depressive symptoms [12,13] and anxiety [13,14]. Chen et al. [15] found that the female gender among adolescents showed a higher risk of depression and anxiety during COVID-19. A study by Majeed et al. [16] added that females reported more somatic complaints, whereas males reported more anger problems. Moreover, 13–15-year-old adolescents were seen to be more depressed than younger children. A longitudinal study in Italy by Conti et al. [14] confirmed that obsessive-compulsive and thought problems among children and adolescents also increased. In addition to that, during the lockdown, family financial hardship was associated with an increase in psychiatric symptoms.

When analyzing behavioral changes and disorders, studies show that the pandemic situation increased children’s and adolescents’ conduct problems and hyperactivity issues [17,18]. There were significant associations between having mental health and conduct behavior [19]. Moreover, non-suicidal self-injury, suicide ideation, suicide plans, and suicide attempts among adolescents in China were higher after the lockdown [20]. Some studies analyzed eating disorders: during the COVID-19 pandemic, 42% of children and adolescents experienced reactivation of eating disorder symptoms despite treatment [21]. Other studies investigated post-traumatic stress disorder (PTSD) and attention-deficit/hyperactivity disorder (ADHD) among children and adolescents during the COVID-19 pandemic. A study by Çetin et al. [22] showed that sleep problems mediated the relationship between PTSD symptoms and severity of ADHD symptoms and the relationship between chronotype and the severity of ADHD symptoms. A literature review by Imran and colleagues [8] revealed that clinginess, distraction, irritability, and fear that family members can contract the deadly disease were the most common behavioral problems identified. Therefore, mental health professionals have found and emphasized many different but mainly negative changes and disorders related to the pandemic experiences among adolescents.

Some studies have approached adolescent health and well-being from the perspective of their parents. Here, Jiao and colleagues, analyzing 320 parents, found that COVID-19 had negative mental health outcomes such as discomfort, agitation, nightmares, fatigue, poor appetite, sleeping disorders, fear for the health of relatives, obsessive request for updates, worry, irritability, inattention, and clinginess [23]. Another study in India [24] revealed that during the pandemic 73% of parents reported that their children were having more signs of increased irritation and 51% of children reported increased signs of anger. According to the parents, these factors were affected by the changes in the child’s diet, sleep, weight, and the increased usage of electronic equipment [24]. Similar findings were found in Spain and Italy, where the most frequent symptoms were difficulty concentrating (77%), boredom (52%), irritability (39%), restlessness (39%), nervousness (38%), feelings of loneliness (31%), uneasiness (30%), and worries (30%) [25]. When comparing adolescents in the general population and with a patient group, a study by Amorim et al. [26] in Portugal showed that 72% of parents reported a change in behavior in children with Autism Spectrum Disorder (ASD) compared to 32% in the control group. Children with ASD and their parents reported higher anxiety levels compared to controls [26]. A qualitative study by O’Sullivan [27] also revealed adverse mental health effects, including feelings of social isolation, depression, anxiety, and increases in maladaptive behavior [27]. Some children suffered from problems that affected their parents, such as financial troubles, or were subject to yelling, shouting, or slapping [8].

Our general overview of the potential impact of COVID-19 on adolescent mental health communicates the researchers, mental health professionals, and parents’ perspectives. However, a more immediate, proximate understanding of adolescents’ direct thoughts, feelings, and perceptions could give additional insights into their experiences of the pandemic in terms of mental health. A better understanding would provide more opportunities to empower the adolescents in helping themselves by promoting certain actions and activities as well as seeking help. Thus, this study aimed to analyze and describe adolescents’ experiences during the COVID-19 pandemic using qualitative design.

## 2. Materials and Methods

### 2.1. Study Design

The study was conducted using a qualitative research design. Semi-structured in-depth interviews were conducted, exploring the experiences of participants and the meanings they attribute to them. The researchers encouraged participants to talk about issues pertinent to the research question by asking open-ended questions, in one-on-one interviews. The aim of this study was to find out the experiences of adolescents during the COVID-19 pandemic.

### 2.2. Sampling and Participants

The interview design was created by this article’s authors (J.L. and K.Š.)—a Ph.D. student and her supervisor. The researchers shared invitations to participate through different social media sites, weblogs, and organizations of adolescents and their parents. Convenient sampling was used. The inclusion criteria of this study were:Age 11–17 years old (grades 5–11);Ability to speak, write, and understand the Lithuanian language;Written consent to participate in the study;Written consents of both parents for an adolescent to participate in the study.

The research sample consisted of 19 adolescents: 10 girls and 9 boys from various urban and rural locations in Lithuania (Table 1). The presence of non-participants was under question for one participant, so his data were excluded from the analyses. This case was not included in the analysis due to an unsafe environment during the interview (his mother was in the same room and refused to leave it).

### 2.3. Data Collection

Data collection took place from 18 April to 22 July in 2021. Due to COVID-19 restrictions, the participants could choose to have an in-person or online interview. The majority of adolescents (11 of 19 participants) preferred in-person interviews. Data were collected using semi-structured in-depth interviews that were audio-taped and transcribed verbatim. The median duration of interviews was 22 min (interquartile range 19–25 min). The interviewers did not personally know nor had a previous relationship with any of the participants or their parents. Locations of interviews were the researcher’s workroom, the private psychologist’s office, or the participant’s home.

Semi-structured in-depth interviews were conducted by one of the co-authors, a Ph.D. student (J.L.), and two other professionals with master’s degrees in psychology (see Acknowledgments). All researchers are practicing psychologists and work at university as researchers with an interest in child and adolescent psychology. The researchers have completed several courses in qualitative research methodology and have at least 5 years of experience in conducting qualitative interviews. The interview consisted of one main question: “*Please, tell me about your mental health during the COVID-19 pandemic*”. Additional questions were: “*What was the hardest part of the pandemic?”, “What helped to overcome the difficulties?*”, and “*What new discoveries has the pandemic brought?*”. After the interview, each participant had an opportunity to share their feelings and, if needed, they were provided with information about the possibilities to seek professional psychological help.

### 2.4. Data Analysis

In this study, a thematic analysis method was applied that is widely used in qualitative research. It is used to analyze classifications and present themes that relate to the data [28,29]. This method implies the coding and categorization of the data into themes. In thematic analyses, processed data can be displayed and classified according to similarities and differences [30].

The thematic analysis provides flexibility for approaching research patterns in two ways, i.e., inductive and deductive [29]. In this study, themes were identified using an inductive approach, which starts with a precise content of interviews and then moves to broader generalizations and finally to themes. This tends to ensure the themes are effectively linked to the data [31]. The data analysis was based on Braun and Clarke’s [28] six phases:*Familiarizing with the data.* Transcribing the data, reading and re-reading the transcript several times, and noting down initial insights and ideas.*Generating initial codes.* Coding important features of the data in a systematic trend across the entire transcripts and collating data relevant to each code. The data were divided into segments according to the changes in the semantic and latent contents.*Searching for themes.* Collating codes into potential themes and gathering all initial codes relevant to each potential theme. For each theme to be included in the results, it had to be mentioned by more than half of the participants (10 or more).*Reviewing themes.* Themes were checked, and changes in the formulation and meaning were made depending on the primary code and the quote. A summary table with codes and related themes was compiled, and then the thematic map was generated.*Defining and naming themes.* Generalizing clear definitions and labeling each theme. The thematic map was validated by qualitative methodology experts.*Producing the report.* Clear, vivid, and the most substantive quotations were selected for the illustrations of themes. The structure of the thematic map was substantiated, and a written report of the analysis was produced [28].

The data analysis was performed by J.L. To avoid personal bias, the initial codes, categories, and themes were reviewed by the supervisor (K.Š.) and qualitative study design consultant (M.K.).

### 2.5. Ethics

Approval for the study was obtained from the Kaunas Regional Ethics Committee for Biomedical Research (No. BE-2-41; 2021-04-04). Participation in the study was anonymous and required the written informed consent of participants and their parents.

## 3. Results

During the qualitative data analysis, five themes were identified and presented in the thematic map (see Figure 1). Each theme is described, interpreted, and analyzed in detail below. For illustration, the quotations of study participants are presented (the brackets indicate the number of an interview and a segment).

### 3.1. Ambivalent Feelings (19/19)

The feelings of the study participants regarding the pandemic period were somewhat ambivalent. In the first stages of the pandemic, some adolescents felt “no effect”. Such adolescents claimed that the COVID-19 pandemic has not affected their normal life or mental health: *“<…> the pandemic hasn’t changed my daily routine much. I’ve been staying at home just as I usually do [smiles]. Not much has changed really”* (Interview No. 4, Segment No. 12). Nevertheless, even the adolescents who initially said that they did not feel any changes confessed further about emotional difficulties: the resulting sadness, feeling of boredom, and wish *“to return to earlier life”* (Interview No. 1, Segment No. 19) without worries about the threat to their own health or even life and those of their relatives.

The interviews revealed that most of the respondents have gone through the pandemic period as a stage triggering mostly negative emotions. The adolescents have mostly talked about such feelings as sadness, anger, loneliness, and boredom. There have been observations that not only mental but also physical health has deteriorated. The feeling of discomfort has been experienced while being in public places, surrounded by people, or communicating physically for a longer period: *“Say, this week we tried to go outside and, honestly, it didn’t feel good to sit in a park full of people, it was disturbing”* (Interview No. 11, Segment No. 12).

It was also observed that the expression of emotions has been changing over time. At the beginning of the lockdown, some adolescents were even happy about imposed restrictions and distance learning from home. They claimed that at the start of the pandemic their mood improved since they did not have to go to school, online lessons finished earlier, and they had more free time for themselves, pursuing their favorite activities: *“<…> that first lockdown—when we did not need to go to school—was cool. Because, well, we practically didn’t do anything”* (Interview No. 1, Segment No. 20). Meanwhile, the others admitted openly that it is very hard to describe their mental health during the pandemic since it has been changing constantly, and there have been ups and downs.

### 3.2. Daily Routine Changes (15/19)

The second highlighted theme is related to the changes in the daily routine during the pandemic. Most adolescents stressed that they can no longer meet with their friends and customarily communicate with them. *“Well, I just missed that possibility to talk to them live, to see them. Maybe to hug them <…>”* (Interview No. 11, Segment No. 13). The participants confessed that they have lacked regular meetings and physical communication not only with their classmates but also with their neighbors, playmates, and the friends they do not see frequently (for example, they go together to the sanatorium every summer). Several participants mentioned that their friends have also changed during this period: the relationship with some of them has been broken; however, new friends have been made. Such changes have been brought about by online communication—new connections have been established during chatting and gaming.

Adolescents often experienced negative emotions due to the inability to entertain, travel, take up something new, or simply continue their usual activities (camps, hikes, scouts, cafés, cinemas, aquaparks, collecting, etc.). They even thought that it may be challenging to get back to the normal contact learning and live communication they had before the pandemic.

When distance education was introduced, the adolescents spent most of the day at home, which in the long run became tiresome. The absence of live meetings triggered a sense of emptiness, and some respondents indicated that their relationship with parents was undermined: *“<…> roughly speaking, for a teenager being closed with parents for a year and a half is not a very good idea. Not the brightest idea that you could come up with <…> The lack of social relationships and, well, all that resulting friction with the parents”* (Interview No. 19, Segment No. 13). The adolescents have lacked privacy and their own space, while that constant tension with the parents has sometimes resulted in disrupted family relations. The research participants mentioned disagreements with their parents and quarrels with their siblings.

Some adolescents also indicated that obligatory wearing of masks was a relatively significant change. They were not used to masks which caused a feeling of discomfort. In their opinion, masks were annoying and uncomfortable; they made them feel bad, and it was difficult to identify other people’s emotions or have quality communication.

### 3.3. Disappointment with Distance Education (10/19)

The theme associated with distance education and related difficulties was emphasized by approximately half of the research participants. In their opinion, this method of learning has been inefficient and has resulted in disappointment. Most participants mentioned the challenge of adapting to the altered form of learning. For some, the learning process was hampered by poor internet connection, while the others found it hard and highly unusual to speak to the screen. They lacked live communication, and various misunderstandings occurred. *“Again, that was a challenge to communicate with new people remotely. Um... and at this point, the impact of the lockdown started; there were some misunderstandings among people. Um, it seemed that maybe it was my fault, maybe it was their fault, and yet, essentially, that was basically the effect of distance learning when you can’t identify properly the emotions of the other person on the screen”* (Interview No. 17, Segment No. 13).

It was also observed that teachers lacked certain competencies to use smart technologies, which might have negatively affected the quality of lessons: *“learning… you still learn less this way. Teachers don’t explain normally, and teachers aren’t skillful with technologies… they also don’t understand a lot of stuff there. There were lots of problems* (Interview No. 5, Segment No. 14). The adolescents reported that sometimes they had to explain to their teachers how to use certain technologies, and at times teachers openly asked more active attendees (e.g., a class representative) for help. Moreover, several adolescents confessed that there was a tendency to finish lessons earlier during distance education.

The study participants highlighted that “<…> it’s much easier to study when there’s this eye contact because a teacher can spend slightly more time next to you. For instance, you can stay after a lesson to learn some more or things like this. It’s way much easier” (Interview No. 14, Segment No. 13). Extracurricular activities—e.g., music lessons—have also been affected. The study participants have noticed that their educational performance worsened during distance education, as it was more difficult for them to concentrate, they got bored or lazy sooner, or they felt demotivated to study: “<…> for some reason I have this strange thing that when I’m at home, I feel very lazy to do anything that’s related. When I went to school, I had this feeling… well… you know, that I was at school, that I had to work here. To study. But when you get home, you do the homework and then ‘wooo.’ You sit [laughs]. You do whatever you want” (Interview No. 4, Segment No. 13).

### 3.4. Coping Strategies (13/19)

When speaking about the challenges experienced during the pandemic, more than half of the adolescents shared with us what has helped them cope with the situation. Mostly, they mentioned their family members or friends. They indicated that communication with other people has brightened their moods. In addition, playing with their pets has had an entertaining and calming effect, when needed. *“My family really helped me… pets. My father is, sort of, really wise, he always has some advice. He speaks a lot and gives advice. So, it comes from him, he doesn’t allow me to get disappointed that much. And my sister, too. She comes and then it’s really cool spending the time with her”* (Interview No. 5, Segment No. 14).

A relatively big part of the adolescents indicated that active leisure was one of the most effective coping strategies. The following physical activities have helped the youngsters to overcome the COVID-19 pandemic and the challenges it has been posing: basketball, a trampoline in the yard, playing on the swings, an outdoor playground, etc. The study participants also mentioned that all such activities have usually taken place in the open air (in their own yard or a park). The research revealed that the boys are inclined to speak more about vigorous physical activities, whereas the girls tend to engage in more creative and quiet activities such as handicrafts, playing a musical instrument, and drawing to let the negative emotions go by transferring them onto paper.

Finally, certain research participants expressed that nothing has really helped them overcome the pandemic challenges, and only the lifting of imposed restrictions has made them feel better: *“Restrictions have been lifted and now I’m feeling great compared to how I was feeling earlier”* (Interview No. 1, segment No. 23). The development of the vaccines has also contributed to the general feeling of well-being: *“Well, nothing actually helped me. It’s just now that the vaccine has become available, I’ve come to realize that we won’t fall ill if we haven’t fallen ill by now. And everything will be all right”* (Interview No. 7, Segment No. 12).

### 3.5. Discoveries (17/19)

Despite the numerous unpleasant experiences, difficulties, and unexpected changes listed above, nearly all participants of our study were able to identify certain discoveries brought about by the pandemic. One of the most frequently mentioned advantages of the COVID-19 pandemic was that more free time has become available. The adolescents tend to use it for their favorite activities that they did not have the time for before the pandemic, e.g., reading historical books, drawing, or sewing. In addition, the participants were very happy to have discovered new activities or pastimes. With the help of smart technologies, adolescents learned new ways of drawing and improved their computer literacy skills. Some research participants managed to discover their favorite music style, created a new style of clothing, or learned skateboarding. Other respondents discovered new places of attraction in the neighborhood during the period since they could not travel abroad like previously.

Some adolescents also spoke about positive changes in their relationships: their relations with family members, classmates, and friends have improved, they have quarreled less, and have learned to appreciate friendship and the people around them more: “*I think I’ve realized how important it is for a human being to have someone around*” (Interview No. 2, Segment No. 14). The respondents confessed that previously they did not attach that much importance to communication until it was restricted: *“We learned that people, a connection is important to us, communication is important. So, I think this has been the greatest kind of discovery for humanity in this period. Because we’ve realized how important each and every one of us is to one another. And it’s not as terrible to communicate as it is to sit in a closed room doing nothing”* (Interview No. 14, Segment No. 14).

During the interviews, the adolescents identified the lockdown as a new experience and a discovery. A number of the research participants also associated this period with personal growth, a better understanding of what they want from life, making plans for the future, greater self-confidence, and better time planning skills. They also mentioned a heightened sense of self-preservation, with the fear of becoming infected with the virus resulting in greater cautiousness. Before the pandemic, the adolescents never considered the importance of self-preservation. Finally, they have come to realize that it can also be fun to be alone or simply to find other alternatives when the usual activities are restricted.

## 4. Discussion

The COVID-19 pandemic affected all humanity. The especially vulnerable group is young people, because they are at a special stage of physical and mental development when social relations and intensive physiological processes play important roles. The hormonal changes that come with puberty collude with adolescent social dynamics to make them highly attuned to social status, peer group, and relationships [8]. Due to the pandemic, many restrictions were introduced which led to school closures, inability to play outdoors, reduced social contacts, altered eating and sleeping habits, etc. Such changes disrupt adolescents’ normal way of life and the fulfillment of their developmental needs, which can cause serious long-term physical and mental health problems [32]. Branquinho et al. [33] argue that listening to adolescents is fundamental in identifying their true needs. Therefore, the aim of this study was to analyze adolescent perspectives on the COVID-19 pandemic experience and its impact on their mental health. After conducting a qualitative study with Lithuanian adolescents, five themes were identified: (1) ambivalent feelings, (2) daily routine changes, (3) disappointment with distance education, (4) coping strategies, and (5) discoveries. These findings are further discussed compared to other researchers and studies.

The ambivalent feelings theme includes mostly negative emotions. The result that COVID-19 affected adolescents negatively has been also demonstrated in other studies. Increased levels of loneliness, depression, anxiety, distress, and decreased life satisfaction are mentioned in the context of the COVID-19 pandemic [11,15,34,35,36,37]. It can also be noted that many studies used the quantitative design with validated scales such as PHQ-9, GAD-7 [13], or DASS-21 [38]. In addition to naming the negative emotions, the adolescents also highlighted some reasons for these emotions to occur. During the interviews, the participants shared that most negative emotions were influenced by their inability to meet peers. The importance of peer communication and social relationships during adolescence is confirmed by previous research, which shows that adolescents are moving towards independence and spend less time with family but more with friends and romantic partners [39].

However, the ambivalence of the feelings was in that adolescents also shared their positive perceptions, mainly related to the early stages of lockdown. Here, the adolescents were happy that they did not need to go to school, they could stay at home, avoid traffic jams while going to school and back, and they did not need to meet in person some people they do not like (peers or teachers). Others were joyful about shorter lessons and less active content in learning activities, which all led to more free time and leisure. We were unable to find other studies with such positive aspects, but from our interviews, it was clear that this positive stage disappeared in a month or two.

Quite expectedly, together with learning changes, the daily routine changes occurred. This encompasses previously common activities such as meetings, trips, after-school events, etc. This is in line with studies by Imran et al. [8] and Rogers et al. [40] who also found that older children and adolescents were disappointed with missing parties, school plays, dance competitions, hanging out with their friends, or sports activities. This may lead to feeling frustrated, nervous, disconnected, nostalgic, and bored because of social distancing during the pandemic. In our study, the most salient daily life change was the restriction of face-to-face meetings, because the study participants were most upset about being unable to meet their peers.

Young people mentioned wearing masks as another new change in their daily lives. Thus far, the studies on the effects of wearing masks are mainly targeted at effectiveness rather than psychological perceptions. The adolescents experienced wearing masks only as negative, presumably due to limited information about its benefits of it in protection against the coronavirus. On the other hand, the young people mentioned that the biggest inconvenience of wearing masks was the inability to see the full face of the other person and the difficulties in recognizing emotions and non-verbal language, which includes eye contact, mimics, and gestures [41]. Facial expressions are essential while eye contact alone is not sufficient in social communication [42].

During the study period, adolescents spent most of their time at home. In such situations, adolescents spent a huge amount of time with family members (from their perspective, sometimes even too much), but still felt a lack of communication with other relatives or friends. Researchers suggest that this could lead to anxiety or depression. A review by Imran et al. [8] also found that the inability to meet grandparents and other relatives was one of the greatest difficulties for older children. Such experiences can be explained by the fact that during adolescence, emotional separation from parents, strong peer identification and socialization with peers, and a sense of community are the main developmental tasks of adolescence [39,43,44]. Other studies showed that the long hours of staying at home (20–24 h) were associated with high levels of anxiety and depression [38].

The systematic review by Pokhrel and Chhetri [45] summarized that online learning, distance, and continuing education have become a panacea for this unprecedented global pandemic, despite the challenges posed to both educators and learners. However, in our study the adolescents experienced only negative experiences—the disappointment with distance education was a very consistent theme. Transitioning from traditional face-to-face learning to online learning can be an entirely different experience for learners and educators, which they must adapt to with little or no other alternatives available. In our study, the adolescents were not fully satisfied with technological opportunities—they were dissatisfied with learning platforms, teachers’ incompetence in distance learning, their hardships with distance learning communication, and moreover, with the inability to meet class friends face-to-face. Finally, this resulted in decreasing academic performance and motivation. Other research on adolescents’ academic achievement or motivation during distance education during a pandemic is ambiguous. The systematic review by Panagouli et al. [46] revealed that students either suffered from learning losses compared to pre-pandemic years or, in some cases, they benefited from online learning. The review showed that most of the students presented a significant decrease in mathematics scores compared to previous years. Younger students faced more difficulties during online learning; however, they presented more enthusiasm for learning materials because they were more creative and interactive [46]. In our interviews, the adolescents did not report specific subjects that would have suffered more due to distance learning than others.

The COVID-19 pandemic increased adolescents’ screen time not only related to distance education. During social isolation, they were exposed to the excessive media coverage of the pandemic [8]. The increase in screen time compared to the pre-quarantine period was found in various countries [32,47,48]. Studies show that excessive screen use in adolescents has been associated with physical and mental health risks. There is evidence that a higher level of screen time is associated with a variety of health harms for children and young people, with evidence strongest for adiposity, unhealthy diet, depressive symptoms, and lower quality of life [49]. Moreover, a systematic review confirmed that social media usage makes children vulnerable to online predators, cyberbullying, and potentially harmful content [8].

Hence, negative daily routine changes and distance education brought quite a lot of negative emotions and feelings to adolescents. Therefore, it was relevant to look at how young people struggled with it and what coping strategies they used. When it comes to coping strategies for dealing with the negative consequences of a pandemic, study participants almost exclusively mentioned communication with family and friends. Adolescents shared that their parents reassured them and provided information about the virus and its management. Young people started to choose to communicate with those friends who know how to cheer them up, with whom it is easy to share feelings, and who had positive thinking. This all had calming effects. The importance of communication on the emotional state has been confirmed by many previous studies. It is important to mention that adolescents did not always use these coping strategies actively and consciously—sometimes it was as a consequence or circumstance rather than an active choice.

Another coping strategy was being with pets. Study participants shared that their mental health was better after playing with their pets. A study by Mueller et al. [50] confirmed that adolescents having pets reported spending more time with their pets during the pandemic, and frequently reported pet interactions as a strategy for coping with stress—they suggested that future research should explore the role of pets in coping with stress and social isolation during the pandemic. Grajfoner et al. [51] also revealed the pets’ positive impact on human mental health and well-being; therefore, they propose the integration of pets in prevention, recovery, and intervention programs to promote mental health and well-being, especially during periods of prolonged social isolation.

The majority of studies analyzing coping strategies during the COVID-19 pandemic were focusing on healthcare professionals, but some of them also addressed adolescent samples. Such studies’ findings were mainly in line with our results. A study by Pigaiani et al. [52] regarding youths aged 15–21 years found that most adolescents planned their daily routine, engaged in structured activities, and developed new interests, and gave a positive reading of the ongoing period, thus revealing adaptive coping strategies. Almost all adolescents kept in contact with their partners, friends, and teachers [52]. Another study showed that 7–18-year-old children and adolescents used various coping strategies to manage psychological changes during the social isolation regarding COVID-19. These strategies were spiritual/emotional (practicing prayer and religious worship; asking for support from people who care about them), cognitive (choosing to receive accurate information from parents and official channels; limiting using of electronic devices), social (spending more time with family; doing at-home activities with the family), and physical (doing more exercise; creating a sleep schedule) [53].

Some participants in our study shared that nothing has really helped them overcome the pandemic challenges. Such a finding could be due to the use of an open-ended question rather than a scale. Since our study participants were quite young, they may have used certain coping strategies without recognizing them as such. For instance, the limitation of information flow could be a coping strategy, but when asking people about this, they may not identify this as a coping behavior. Previous studies found that self-control enables children and adolescents to cope with the experience of emotional challenges or difficult situations [54]. A qualitative study of Portuguese adolescents’ and young people’s experiences during COVID-19 also found that the most common coping strategies were regular communication with family and friends via video calls, doing pleasurable activities, leading a calm and positive life, and having routine and scheduled time [33]. All these results are in line with our study, and it reveals that most adolescents have tried to help themselves, but sometimes they needed help or support from their relatives or significant others. Therefore, both personal and environmental coping resources are relevant to adolescent mental health and thus should be considered for prevention and early intervention.

Across uncertainties and negative and ambivalent experiences, one clearly positive theme emerged—the discoveries. In our study, almost all participants shared discoveries ranging from fun activities to personal growth. Another qualitative study with adolescents and young people (16–24 years) also confirmed that the pandemic helped to understand the importance of relationships and helped to strengthen and select friendly relationships. During the pandemic, there was more time for pleasurable activities and thinking and setting goals; some students started exercising, enjoyed time with parents, and had opportunities for personal growth [33]. These results are quite similar to our study, and to our knowledge, more studies analyzing such adolescent experiences during COVID-19 have not been found. A qualitative study by Todorova et al. [55] with adults (mean age 37 years) revealed that even adults during the pandemic rediscovered themselves, turned inward and underwent personal growth, or wished to make personal changes, such as: new or revived health-promoting behaviors (going to the gym, losing weight, retaining new hygiene habits, or changing eating habits), improving skills (e.g., attending professional workshops), changing financial behaviors (saving in case of a new crisis), and using time differently (relaxing, maintaining a routine) [55]. We suggest that the theme of discoveries emerged mainly due to the qualitative open-ended design because the quantitative studies on mental health during COVID-19 were mainly addressing negative aspects of health, well-being, and behaviors. From our perspective, this is a valid finding because almost all adolescents in our study were keenly sharing their positive insights.

### 4.1. Strengths and Limitations

There are several strengths and limitations to be considered in this study. One of the advantages of this study is that the problem was analyzed from a qualitative point of view. The qualitative design allowed to evaluate not only the manifestation of certain symptoms but at the same time to find out the causes of certain phenomena that occurred (for example, why adolescents were happy about the distance learning at the beginning of the lockdown). It was also possible to discover new approaches, experiences, and unexpected themes that might not have been possible with pre-constructed questions and limited response options. Another advantage of our study could be that the sample of the study was quite heterogeneous: the study included young people of different ages (from 11 to 17 years old) and was well-balanced by gender (10 girls and 9 boys). This study also involved adolescents from various schools and municipalities across Lithuania (rural and urban residence), which provides experiences of living in different contexts.

The major limitation of this study is the small study sample, and therefore the conclusions should be generalized with caution. Though a small study sample was the basis for a thorough analysis of individual experiences, larger-scale representative studies would allow the validity of these qualitative analysis findings to be verified in the population. We had no objective data on the mental health of the study participants before the COVID-19, so not every finding could be imputed to the pandemic. In addition, only volunteers from the general population participated in the study, so our findings may be less representative of adolescents who were shy, less reflective, or had predominantly negative moods. Only a certain group of motivated, reflective, or internally conflicted participants are likely to respond invitation to participate. However, a group that never participates in the research is out of reach. In this study, there were no adolescents not attending school, so it is hard to estimate how much our findings are consistent with their perceptions of mental health.

Moreover, during the lockdown, the lack of social opportunities may have had a negative impact on social skills [56], so some young people may have skipped the invitation to study just based on their reluctance to communicate. These concerns support the limited extrapolation of our study findings; for instance, there is evidence in the literature that children and adolescents with physical or mental disabilities survived the COVID-19 pandemic even harder [26,57]. Finally, the way additional questions were formulated may also have affected some study findings, even though not every interview needed these specific additional questions.

### 4.2. Recommendations

All efforts to improve mental health during the pandemic restrictions should be mainly targeted at the educational environment because participants mostly reflected on mental health being affected by educational factors related to distance learning. Therefore, based on this study’s findings, we think that some suggestions for policymakers could be elaborated. First, the schools’ closures should take place as a last resort during the pandemic, i.e., closures would take place only after all other preventive measures have been introduced. In addition, the shifting education could be introduced during the pandemic: for instance, half of the classes (groups) learn face-to-face for two weeks, then shift to online and leave the live learning in schools for another half of the students. Third, there is an increasing need for psychological skills such as self-efficacy, stress management, emotional regulation, team building, communication, or resilience—and not only during a pandemic but in regular times as well. Such skills could be developed by adding them to curricula instead of some subject classes. Finally, the classes for development of psychological skills could be additionally built-up during the lockdowns and distant learning weeks. In addition, it should be kept in mind that the psychological skills development is more effective with smaller groups, for instance up to 8–12, which makes the need for more human resources at schools.

## 5. Conclusions

The COVID-19 pandemic affected not only physical but also mental health. The social distancing, lockdown, school from home, and work from home became commonplace. This study regarding adolescents revealed their mostly negative feelings such as sadness, anger, loneliness, or boredom. The study participants shared about frustrating daily routine changes (mainly loss of favorite activities due to restrictions) and shifting to distance education mode that was quite limited in effectiveness and convenience. Nonetheless, communication with family, peers, pets, active leisure, and favorite activities helped them to cope with the difficulties during COVID-19. During this period, adolescents had more free time for themselves and personal growth, found new activities, or improved some relationships. Overall, the COVID-19 pandemic had mostly negative side effects, and regardless of positive experiences was mainly considered by adolescents as adversity for their mental health.

## Figures and Tables

**Figure 1 ijerph-19-07086-f001:**
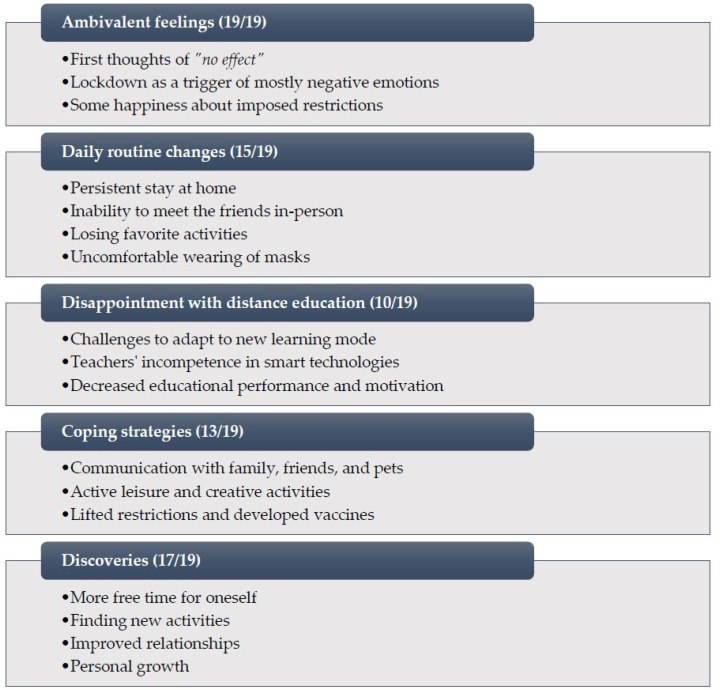
Thematic map: COVID-19 experience among adolescents.

**Table 1 ijerph-19-07086-t001:** The main characteristics of the study sample.

No.	Name (Changed)	Gender	Age (Years)	Grade	Mode of Interview
1	Urtė	Girl	17	11	In-person
2	Silvija	Girl	14	8	Online
3	Eglė	Girl	11	4	In-person
4	Sigis	Boy	17	11	In-person
5	Jonas	Boy	12	6	In-person
6	Tomas	Boy	11	4	In-person
7	Emanuelis	Boy	12	5	Online
8	Vilius	Boy	12	5	Online
9	Saulė	Girl	14	8	In-person
10	Petras	Boy	15	8	Online
11	Vaiva	Girl	12	5	Online
12	Tadas	Boy	17	10	In-person
13	Augustas	Boy	11	4	In-person
14	Miglė	Girl	14	8	In-person
15	Gabija	Girl	12	6	Online
16	Paulius	Boy	14	6	In-person
17	Danutė	Girl	16	9	Online
18	Laura	Girl	15	8	Online
19	Greta	Girl	17	11	In-person

## Data Availability

Not applicable.

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
