# Peer review of "Mental Health during COVID-19 Pandemic: Qualitative Perceptions among Lithuanian Adolescents"

_ijerph, 2022, doi:10.3390/ijerph19127086_

Round 1

Reviewer 1 Report

The study aims to explore an interesting topic: Adolescents’ experiences during the COVID-19 pandemic.  

Overall, the paper is clearly written, sections of the paper (Introduction Methods Results Conclusion) are properly formatted in abstract and text. Some specific comments are listed below to assist the authors to improve the manuscript.

Discretionary Revisions

A study of this nature would have benefited from more qualitative approaches, such as a systematic comparison of the study data with those from international and regional ones. In fact the ms could be strengthened with a systematic review to be undertaken aiming to improve the comparison of these results with those of previous or similar studies and briefly explain why results may be different. In general, the comparison of these results with those of previous studies should be improved.

Methods: Given the study design and the type of data used, the authors should clarify for readers except for their identity, their occupation, experience and credentials. Subsequently this improves the credibility of the findings by giving readers the ability to assess how these factors might have influenced the researchers’ observations and interpretations. Since qualitative researchers closely engage with the research process and participants, how have the authors checked for the reliability of replies in order to avoid personal bias?

Participant selection: Authors should report reasons for non-participation. Convenience sampling is less optimal because it may fail to capture important perspectives from difficult to reach people. Rigorous attempts to recruit participants and reasons for non-participation should be stated to reduce the likelihood of making unsupported statements.

Authors should describe presence of non-participants during interviews because it illuminates why participants responded in a particular way. For instance, participants might be more reserved and feel disempowered talking on sensitive topics if their parents are present.

The authors should be more precise on assessing  the consistency between the data presented and the study findings, including the both major and minor themes.

In conclusion, the authors have made an intensive effort to exploit and present data of Lithuanian adolescents’ experiences during the COVID-19 pandemic.

Author Response

Please, find the attached document.

Reviewer 2 Report

I find this paper focused on adolescent experience during the COVID-19 is important and interesting. I only have some minor questions.

1. The title of the paper focuses on mental health during the COVID-19 pandemic, but I find it may beyond mental health, such as the five themes included daily routine changes which may differ with mental health.
2. One main question used in the interview is “Please, tell me about your mental health during the COVID-19 pandemic”. I’m wondering whether adolescents would have the same idea about mental health. For example, some may believe it may be mainly related to negative emotions, but others may not. 
3.    Do the authors find something specific for the adolescent during the COVID-19? They may discuss more the results that are particularly relevant to the adolescent.

Author Response

Please, find the attached document.

Reviewer 3 Report

First of all, I wish to congratulate the authors for conducting such a pertinent and important study in a timely fashion. This study aims to investigate Lithuanian adolescents’ mental health against the background of the COVID-19 pandemic via the lens of in-depth qualitative interviews. Overall, this study is of high impact and could help enrich the literature and inform better policymaking. Please find my comments below and address them properly, as I believe that, while they are minor in nature, shedding light on these concerns could help the authors further strengthen their work and the readers better appreciate the study.

The authors mentioned that five themes were identified, “Ambivalent feelings; Daily routine changes; Disappointment with distance education; Coping strategies; Discoveries”. Noticeably, the topic of education stands out, as not only it is represented in all five themes, but also it is highlighted via the “Disappointment with distance education” theme. Could the authors please elaborate on this issue in the manuscript to further shed light on the unique role education played with regard to the participants’ mental health?

A related concern I have centers on the characteristics of the participants. Were all of them enrolled in school at the time of the interview? What percentages of Lithuania youths are in school vs. out of school? While finding replicability is less of an issue in qualitative studies, please consider shedding some light on the issue to further help the audience understand the depth/complexity of the research topic.

Could the authors please consider defining the five themes that emerged from the analyses? Right now, it is unclear in terms of what they entail. For instance, rather than “ambivalent feelings”, based on what is being described between lines 191-216, it seems that the participants’ attitudes/emotions toward the pandemic are progressing (e.g., “…confessed further about emotional difficulties…” & “…but from our interviews, it was clear that this positive stage disappeared in a month or two”).  

Based on the research findings, what concrete suggestions would the authors provide to policymakers to help adolescents better cope with their mental health challenges amid the pandemic and beyond? Please consider elaborating on these suggestions to further strengthen the manuscript.  

Author Response

Please, find the attached document.

Reviewer 4 Report

Dear authors,

please thank you for allowing me to review your compelling article. I think with some adjustments it can be improved and published.

- Firstly, I think the title can be made more comprehensible. Your research question is not about mental health, but about experiences. I would suggest something like "A qualitative study of Lithuanian adolescents' experiences during the COVID-19 pandemic."

- I think the abstract in grammar and speech should be improved.

- I would recommend reviewing the opening sentence of the introduction section as it does not show pandemic declared by WHO.

- Your manuscript is very weak methodologically.

- I would suggest structuring the methods section by inserting paragraphs and expanding the content as follows:

Study design

Sampling and participants

Data collection (Since it involves under-age participants, it is interesting to have data on how recruitment took place).

Data analysis (in this section describe your research process, even describing who did what, not only reporting what Braun & Clarke suggested doing a thematic analysis).

Ethical issues

Inside these subparagraphs, transparently explain the research process. For clearly reporting the research process, I suggest authors following the COREQ (COnsolidated criteria for REporting Qualitative research) Checklist (https://www.equator-network.org/reporting-guidelines/coreq/).

- As I read, you conducted in-depth interviews, and not semi-structured as you stated. I would suggest providing the full interview guide, with the topics addressed which you decided on beforehand, or better specify the type of data collection instrument you chose.

- In addition, results from process (number of participants, interview duration, interview setting choice and rationale) should be placed in a paragraph called "Results." While results derived from the data should be entered in a paragraph called "Findings."

- The discussion section is very interesting and well structured. Providing suggestions on adolescents' perceptions and spillovers regarding decisions made by policy makers during the pandemic period would be useful.

Author Response

Please, find the attached document.

Round 2

Reviewer 3 Report

Congratulations and well done. My recommendation is "Accept".

Author Response

Thank you for Tourcomments. We appreciate that.

Reviewer 4 Report

Dear authors,

as for the title, I would suggest harmonizing it with your study findings, and not with your "a priori" investigative intentions.

The method has improved greatly by following the COREQ Checklist. However, I would like to raise a few points:

Regarding the interviews you conducted:

- first, I reiterate that such a unstructured guide is methodologically an in-depth interview. Also, please remove any reference to words such as "questionnaire."

- secondly, the additional questions seem to come from biases and pre-assumptions of the researchers.

For these two reasons, I suggest reflecting on the type of data collection instrument adopted.

Regarding the suggestion to name a paragraph devoted to the analysis's own results as a "findings section," as well as regarding data presentation, I think it is a matter of expertise in qualitative research rather than adherence to examples and guidelines.

Best regards

Author Response

Thank you for your comments. We appreciate that. 

Also, we discussed the data collection instrument and we decided to amend the manuscript. The semi-structured in-depth interview method was applied in our study. We changed it in the whole manuscript.

We remove the references to the words "questionnaire." Thank you for your attentiveness.

And finally, we added a few sentences in the Discussion section (Limitations) according to your comment on biases and pre-assumptions.